# Changes in Package Sizes of Savoury Snacks through Exploration of Euromonitor and Industry Perspectives

**DOI:** 10.3390/ijerph19159359

**Published:** 2022-07-30

**Authors:** Hei Man Emily Ng, Jessica Xu, Qingzhou Liu, Anna Rangan

**Affiliations:** 1Discipline of Nutrition and Dietetics, Susan Wakil School of Nursing and Midwifery, Faculty of Medicine and Health, The University of Sydney, Camperdown, NSW 2006, Australia; heng0680@uni.sydney.edu.au (H.M.E.N.); jexu3507@uni.sydney.edu.au (J.X.); 2Charles Perkins Centre, The University of Sydney, Sydney, NSW 2006, Australia; qingzhou.liu@sydney.edu.au; 3School of Life and Environmental Sciences, Faculty of Science, The University of Sydney, Sydney, NSW 2006, Australia

**Keywords:** package size, savoury snacks, packaged snack size, food industry

## Abstract

Portion sizes of many energy-dense and nutrient-poor foods and drinks have increased in the past decade, whereas our understanding of the pattern of changes in package sizes remains limited. This study aimed to determine changing trends in sales and package sizes of savoury snacks in Australia, the USA, Japan and Hong Kong, and to investigate industry perspectives for these changes. Sales data (units per capita) between 2006–2020 on savoury snacks were extracted from the Euromonitor International database. Industry perspectives on package size changes were extracted systematically from selected databases, company reports and related websites following the PRISMA-ScR guidelines. The findings showed that sales per capita of savoury snacks of all package sizes increased across all four countries/regions between 2006–2020. Although changes in the proportion of smaller (<100 g) versus larger (>100 g) package size sales in each country/region over time were modest, Japan and Hong Kong exhibited a consistently higher proportion of smaller package sales compared with Australia and the USA (83.3%, 64.4%, 44.3%, 20.2%, respectively). Industry perspectives showed that increasing consumer health consciousness, demands for convenience and portion control were the main contributors to decreasing package sizes of savoury snacks. Industry reports from 2020 showed an increase in larger package size sales due to consumer purchasing behaviour amidst the COVID-19 pandemic.

## 1. Introduction

Portion sizes of many foods and beverages have increased over the past decade, especially those of energy-dense, nutrient-poor discretionary foods [1,2] linked to the current obesity epidemic [3]. It has been shown that offering larger servings of foods and beverages leads to higher energy intakes without compensatory reductions at subsequent meals [4]. This ‘portion size effect’ is particularly evident with packaged snacks, whereby energy intake has been shown to increase with larger package sizes [5]. 

A 2018 study by Wang et al. found that the consumption of energy-dense, nutrient-poor snacks differed between Western and Asian countries with the prevalence of snacking and the proportion of total daily energy intake from snacking being higher in Western countries (Australia, the USA) compared with China [6]. In Australia and the USA, snacking contributed around 30% and 24%, respectively, of total daily energy intake amongst 2–18-year-old children, while China’s estimated energy intake from snacking remained relatively low at 6–8% [6]. Similar studies among adults are lacking but estimates of snacking in young children under seven years old in Japan [7] and Hong Kong [8] approximate 20–22% of total energy intake.

In both Western and Asian countries, overall snack consumption has increased with notable growth in the savoury snack category [9]. The global market size for savoury snacks is expected to rise from 67 billion USD in 2020 to 102 billion USD by 2028 [10]. A 2020 study from independent research company Roy Morgan revealed that 90% of Australian adults consumed packaged snack food in an average week, with “savoury snacks” being the most popular category, consumed by more than 66% in a week [11]. In addition, a 2021 report by global data platform Statista revealed that savoury snacks were consumed in more than 90% of US households per week, with two out of three households routinely consuming at least three types of savoury snacks per week [12]. In Japan and Hong Kong, savoury snack consumption increased steadily over the past decade due to the growing influence of Western snacks and their widespread availability [13,14]. An international survey conducted in 2020 found that the COVID-19 pandemic had accelerated global levels of snacking, with 46% of respondents snacking more frequently during the pandemic as opposed to prior [15].

With an increase in snacking behaviour and upsizing of packages over time, it is timely to explore the sales data of different sized snack packages. In this study we assessed the trends in savoury snack packaging sizes using sales data available on Euromonitor. Two Western (Australia, the USA) and two Asian (Japan, Hong Kong) countries/regions were chosen to investigate savoury snack consumption amongst different cultural populations. Australia, the USA and Japan are mass producers of savoury snacks while Hong Kong was chosen due to its similar urbanised lifestyle to the other areas of interest and as a major importer of both Western and Asian savoury snacks. Analysing how package sizing of savoury snacks has changed over time will allow a better understanding of the food environment and its influence on consumers’ food choices and eating behaviours [16,17]. Gaining an understanding of the industry perspectives, consumer demands and current sales trends can help guide pragmatic public health policy. Therefore, this study aimed to determine changes in sales of different package sizes of savoury snacks between 2006 and 2020 in Australia, USA, Japan and Hong Kong, as well as to investigate reported industry and marketing perspectives for these changes.

## 2. Methods

### 2.1. Trends in Sales Data of Savoury Snacks According to Package Size

Sales data for savoury snack packaged foods were sourced from the Euromonitor Passport Global Market Information Database, 2021 Edition (Euromonitor, London, UK). The Euromonitor database collates data from numerous international primary and secondary sources, such as company financial reports, store audits and government data, to provide statistics, analyses, reports, surveys and breaking news on industries, countries and consumers worldwide [9].

#### 2.1.1. Definitions and Data Collection for Savoury Snack Package Sizes

Sales data for units of packages were obtained from the years 2006 to 2020 for four areas of interest: Australia, the USA, Japan and Hong Kong, using a full tree search from Euromonitor (Appendix A). Sales data of savoury snacks, including packaging unit volume of sales and compound annual growth rate (CAGR), were exported from Euromonitor into Microsoft Excel. Only retail sales data (excluding food service data) were available for the savoury snacks category, defined as sales through establishments primarily engaged in the sale of goods for home use and consumption (Appendix A). The ‘savoury snack’ category on Euromonitor is well defined and divided into six subcategories: Nuts, seeds and trail mixes; Salty snacks; Savoury biscuits; Popcorn; Pretzels; and Other (Appendix A). CAGR is the annual average growth rate expressed in percentage terms for a specified forecast period [9]. Packaging unit volume data of savoury snacks pertained to the number of packaging units sold to the consumer through all retail channels [9].

#### 2.1.2. Data Analysis and Graphing

Package size data were organised into size bands and year bands to examine the trends over time. The four size bands (0–50 g, 51–100 g, 101–300 g, >300 g) for savoury snacks were chosen for consistency and efficient data analysis in alignment with the existing products in the market. Upon market analysis, packages weighing <50 g are often labelled as fun-sized portions or individual packages from a larger multipack, while 51–100 g packs are usually single serves. Packages weighing between 101–300 g tend to be labelled as family packs, and those >300 g are likely to be party-size share bags. The 15-year period was similarly divided into five three-year bands of 2006–2008 (earliest time at which data were available), 2009–2011, 2012–2014, 2015–2017 and 2018–2020. The package unit sales data were converted into sales per capita using population statistics for each respective country/region and year available on Euromonitor. Package unit sales per capita were then tabled and graphed as absolute values and by proportion of total sales per capita with the size bands and year bands outlined for each country/region. CAGR per capita was charted to visualise the changes in sales of different package sizes of savoury snacks over time.

### 2.2. Food Industry Perspectives on Changing Savoury Snack Package Sizes

To explore industry perspectives on changes in savoury snack sales and package sizing over time, database and grey literature searches were conducted in accordance with the PRISMA-ScR (The Preferred Reporting Items for Systematic Reviews and Meta-Analyses Extension for Scoping Reviews) guidelines [18]. 

#### 2.2.1. Inclusion and Exclusion Criteria for Search Strategy

Eligible sources included official reports, company financial reports, journal or newspaper articles that contained information about changes in savoury snack packaging sizes in Australia, the USA, Japan or Hong Kong, and published between 2006–2021. Sources were excluded if they were not available in the English language, were not related to the four chosen areas of interest, or did not clearly and specifically describe changes in package sizes relating to savoury snacks.

#### 2.2.2. Search and Screening Strategy for Articles, Reports and Other Literature

Systematic keyword searches were performed on two selected databases (Medline and Proquest Central). Industry reports from the Euromonitor database were extracted using a similar search tree strategy as the one for obtaining sales statistics (Appendix A). A grey literature search was conducted using a search engine (Google) to further source any relevant grey literature. As per Dumas et al. [10], the first 100 search results from Google were chosen to be analysed. The full search strategy is attached (Appendix A).

All searches were conducted on 6 October 2021. The data selection process was performed by two reviewers independently using the web-based reference management platform, Covidence (2021 edition). The screening was a two-part process beginning with title and abstract review, followed by full-text appraisal. Data extraction was conducted by both reviewers collaboratively. Any discrepancies in judgement were discussed with a third reviewer. 

## 3. Results

### 3.1. Trends in Sales Sata of Savoury Snacks According to Package Size

#### 3.1.1. Total Sales per Capita

Total sales per capita of savoury snacks increased between 2006 to 2020 in Australia, the USA, Japan, and Hong Kong (Table 1). Hong Kong’s total sales per capita were markedly lower compared with the other three areas of interest. 

#### 3.1.2. Proportion of Total Sales per Capita by Package Size Band

As a proportion of total sales per capita, the largest package size band (>300 g) decreased from 2006 to 2020, while the smallest package size band (<50 g) increased in all four countries/regions (Figure 1). 

The most popular package sizes, in terms of proportion of total savoury snack sales per capita, varied between areas. The 0–50 g and 51–100 g size bands were equally popular in Japan while Hong Kong sales were highest for the 51–100 g package size band. The 101–300 g size band was the most popular band in Australia and the USA. The two smaller package size bands (0–50 g and 51–100 g) were more popular in Japan and Hong Kong, contributing on average 83.32% and 66.36% of savoury snack sales, respectively, compared with 44.35% in Australia and 20.16% in the USA. The USA exhibited a much higher percentage of savoury snack sales from the largest >300 g size band, averaging 34.57% of total sales across the years, compared to in Australia (2.11%), Hong Kong (1.53%) and Japan (0.02%).

#### 3.1.3. CAGR of Different Package Size Bands of Savoury Snacks

CAGR per capita from 2006 to 2020 was positive across all savoury snack package size bands in Australia, the USA and Hong Kong, with the exception of Japan (−2.67% CAGR in the >300 g size band) (Figure 2). Across the four areas of interest, the two smaller package size bands (0–50 g, 51–100 g) had greater growth cumulatively compared with the two larger size bands (101–300 g, >300 g). The largest CAGR per capita increase for each country/region varied, with Australia and the USA showing the highest growth in 51–100 g packages (+3.06% and +2.65%, respectively) while Japan and Hong Kong had the highest growth in the 0–50 g size band (+2.90% and +2.72%, respectively). 

#### 3.1.4. CAGR per Capita of Different Categories of Savoury Snacks 

The CAGR per capita for each snack category, as defined by Euromonitor, varied among areas of interest between the years 2016–2021 (Appendix A). In Australia, the category with the highest CAGR per capita was Popcorn at 46.7% followed by Rice snacks (25.8%), whereas in the USA, the highest growth categories were Tortilla chips (21.5%) and Puffed snacks (18.9%). The highest CAGR category in Japan was Potato chips (11.3%) followed by Nuts, seeds and trail mixes (8.6%), while CAGR in Hong Kong showed sizeable growth in the Vegetable, pulse and bread chips category (55.9%) followed by Nuts, seeds and trail mixes (11.5%). 

### 3.2. Food Industry Perspectives on Changing Savoury Snack Package Sizes

#### 3.2.1. PRISMA Search and Screening Results

The initial electronic database and grey literature search identified 868 records of potential interest. After removing duplicates, 642 records remained for the title and abstract screening (Figure 3). After excluding 586 records, 56 full-text records were assessed against the eligibility criteria. A total of 34 records were included for data synthesis. Most of the included records originated from the USA (*n* = 21) followed by Australia (*n* = 5), global (*n* = 4), Japan (*n* = 3) and Hong Kong (*n* = 1). The majority of articles were published between 2015 to 2021.

#### 3.2.2. Food Industry Perspectives for Changes in Savoury Snack Package Sizes

All included records were grouped according to the direction of package size change and the main reasons driving these changes (Table 2; see Appendix A for more detail). A total of 25 studies reported on a decrease in package size while nine sources reported an increase.

Health consciousness was the most common reason cited for reducing package size (*n* = 15), followed by convenience (*n* = 9), portion control (*n* = 8), and other motives (*n* = 4) such as product line expansion and increasing profit. The main drivers behind increasing package sizes included COVID-19 snacking behaviour (*n* = 6), perceived value for money (*n* = 5), and social sharing behaviour (*n* = 3). 

## 4. Discussion

Using a systematic methodology, this study explored the changing trends in total and package size sales of savoury snacks over a 15-year period (2006–2020) and investigated industry perspectives for these changes across Australia, the USA, Japan and Hong Kong. Total unit sales per capita of savoury snacks increased across all areas of interest over time, illustrating the continuing rise in popularity of the savoury snack category globally. Positive CAGR was found in most package sizes across all four areas of interest. Smaller package sizes (<100 g) showed higher growth rates compared with larger package sizes (>100 g) during this time period. Industry reports mirrored this downsizing trend, citing three main factors as increased health consciousness, demands for portion control and convenience. Although the majority of industry reports recorded a decrease in package size, more recent sources (2020–2021) revealed a resurgence in sales of larger package sizes due to changes in snacking behaviour during the COVID-19 pandemic.

Australia and Japan showed the highest overall number of savoury snack sales per capita followed by the USA and Hong Kong. Studies on the prevalence and frequency of snacking across different regions may help to explain this trend. In Australia, almost 90% of adults and 96% of children consume packaged snack food in an average week [11,19]. In Japan, a 2020 survey conducted on consumers (10–79 years) showed that 71% consumed snacks weekly, with most respondents falling into the once-a-day category [20]. In the USA, the prevalence of snacking was 97% in adults and 94% in children [21], yet the USA ranked only third out of four in the overall number of savoury sales per capita in this study. This could be due to the USA showing a larger proportion (approximately 35%) of savoury snack purchases in the largest size band (>300 g) compared with the other three countries/areas of interest. This suggests that US consumers may be purchasing fewer individual packages of snacks and likely consume one large package over multiple snacking occasions [21,22]. This can be contrasted to Japan which ranked second out of four in overall number of savoury snack sales per capita but only had 0.02% of total sales in the >300 g size band. In China, including the special administrative region of Hong Kong, prevalence of snacking was on average 53% in children and 35% in adults per week; the lowest of the four areas of interest [21]. Hong Kong’s markedly lower total sales per capita may be due to Chinese consumers’ preferences for fruits, beverages and sweet grain products such as rice cakes as the most favoured types of snacks [21]. The savoury snack category was the most popular snack category in Australia and the USA, but one of the least preferred in China [11,23]. 

Smaller package sizes (<100 g) accounted for a higher proportion of total savoury snack sales in Japan and Hong Kong, compared with Australia and the USA. Although reasons for this contrast are not clear, it can be speculated that the rapid increase in single-person households in Asia as well as smaller living spaces compared with Western countries may contribute to this difference [24,25]. The rise in single-person households would likely result in less incentive to purchase larger ‘sharing’ or ‘family-sized’ packages of snacks. Furthermore, living quarters in Asian countries are, on average, smaller than living spaces in Western countries due to the high population density [25], and thus less storage space for larger packages of snacks. Market analysis in 2021 shows that Japanese snacks, in which Hong Kong is a large importer of, are generally smaller in package size compared to Western counterparts [26,27,28]. 

Prior to the COVID-19 pandemic, industry perspectives showed a consistent shift towards smaller package sizing of savoury snacks, mainly attributed to increasing health consciousness amongst consumers and demands for convenience and portion control. A survey on food priorities revealed that consuming a smaller portion size of food was one of the top three priorities amongst one-third of Australians surveyed in 2015 [29]. In 2019, over 16% of revenue for Mondelez International, the second largest savoury snack producer in the USA, came from smaller packages sold for portion control purposes. Furthermore, Mondelez forecasted that 20% of global snacks net revenue will come from portion-controlled products by 2025 due to increasing health awareness and consumers leading busier “on-the-go” lifestyles [29]. Indeed, health consciousness trends can be seen not only in the increasing sales of smaller package sizes but also in the growth of different categories of savoury snacks. For example, non-deep-fried vegetable pulse snacks contain approximately 490 calories, 2.5 g saturated fat and 260 mg sodium per 100 g [30], while the same quantity of chips contain approximately 570 calories, 5 g saturated fat and 600 mg sodium per 100 g [31]. Australia had the largest growth in the popcorn and rice snacks categories while Hong Kong showed a marked increase in the vegetable, pulse and bread chips category from 2006 to 2020, illustrating consumers’ shift towards ‘healthier’ snacking options globally [32,33]. 

The shift towards smaller package sizes may also be attributed to the eating behaviours of the millennial age group, where millennials have demonstrated a preference for single-portion snack foods over traditional sit-down meals [24]. Forty percent of Americans report replacing meals, especially lunch, with snacking [34], whilst in Australia, 70% of millennials preferred snacking throughout the day rather than consuming more traditional meals [35]. Asian countries have also shown an increase of 12% in snacking throughout the day in 2015 [36]. Manufacturing companies reported being responsive to consumer demands by producing smaller package sizes of snacks. For instance, Mondelez International aimed to provide less than 200 calories per single package to support consumers increased health consciousness, busy lifestyles and preferences for snacking as meal replacements [37]. These findings are consistent with those studies examining sales of package sizes of carbonates, that reported growth in sales of smaller package sizes and a reduction in larger sizes of carbonates over the past decade [16]. 

Other industry reasons reported for the reduction in package sizes revolved around costs and the desire to expand product lines by introducing smaller packages of new items for consumers to trial [38,39,40,41,42]. In Asia, Calbee introduced the 50 g pack single serving and smaller sized packs to attract domestic consumers to sample new products [41]. Smiths Snackfoods, the largest savoury snack producer in Australia, reported the need to shrink package sizes to make up for higher manufacturing costs [39]. For example, Red Rock Deli potato chips fell from 185 g to 165 g in 2014 and Smith’s Potato Chips and Doritos Corn Chips dropped from 200 g to 175 g in 2010 and has since further decreased to 170 g upon market investigation in 2021 [39]. This package size reduction occurred without a concomitant drop in the unit price of the product, colloquially referred to as ‘shrinkflation’, and is a common method employed by manufacturing companies to increase profits [43].

The COVID-19 pandemic in 2020–2021 brought a resurgence in sales and demand for larger package sizes and multipacks of savoury snacks according to industry research [44] as movement restrictions presented by lockdowns worldwide led to more indulgent snacking behaviours at home [45]. Mondelez International’s annual State of Snacking study revealed that 52% of adults globally found snacking acted as a “lifeline” throughout the pandemic, with 88% of adults snacking more or the same during COVID-19 compared to previously [44]. Furthermore, social interactions within households, including snack-sharing behaviour, were highly valued amidst the pandemic [46]. Global lockdowns may have resulted in consumers bulk-buying groceries, leading to the purchasing of larger packages to minimise separate supermarket visits [47]. Perceived value for money was another apparent reason for consumers purchasing larger package sizes, as a lower price per unit appears to be better ‘value for money’ [48]. 

There has been compelling evidence in the literature addressing the ‘portion size effect’, showing that offering larger portion sizes of discretionary foods and drinks increases the amount consumed [2,49]. Reducing the size of a single package has been found to be a promising strategy to counter this effect as consumers are more likely to exercise self-restraint and consume smaller amounts of food when offered foods in smaller servings or packages [48,50]. Meta-analysis and modelling indicate that eliminating larger portion sizes from the diet could reduce the average daily energy intake of US adults by 22–29% [49]. However, the effectiveness of this strategy in real-world settings has yet to be established [49]. Overt actions to reduce portions by making package sizes smaller encountered consumer resistance, especially without a linear price drop [51]. Hence, support from industry and government bodies is needed to normalise smaller package sizes in the market and recalibrate consumers’ perceptions of normal portion sizes [52].

Public health initiatives that have led to legislative regulation enforcement in reducing savoury snack package size and consumption are currently limited in the four countries/areas of interest. A search for public health campaigns to reduce portion sizes of unhealthy foods and drinks did not reveal any results for Japan and Hong Kong. In Australia, the Healthy Food Partnership, consisting of food industry and public health representatives, undertook an evidence and policy analysis to assess the effectiveness of potential strategies to achieve optimal portion control. This report found that restricting portion sizes of discretionary foods and drinks in public and private sector settings were potentially effective if accompanied by a reduction in prices, given that perceived value for money is an important consideration for consumers [49]. The Partnership is currently developing a best-practice guide that will provide guidance and support to the food industry to promote and provide healthier serving sizes [53]. A Delphi study conducted by the Portion Balance Coalition in the USA showed that stealth interventions such as creating an artificial stopping point in packages may be the best method to reduce food consumption without inducing consumer resistance [54]. However, the effectiveness of this method has not yet been investigated in real-life settings. 

Other public health initiatives that have shown some success in regulating industry and food manufacturing companies to help consumers decrease consumption of savoury snack foods include the UK government’s Public Health Responsibility Deal calorie reduction pledge which encouraged food businesses to help consumers decrease energy intake with portion size reduction as one potential avenue [51]. Manufacturing companies were more likely to voluntarily participate due to corporate social responsibility and reputational enhancement [55]. Similar government initiatives could be adopted to encourage food manufacturing companies to decrease package sizing, which in turn, create a better food environment.

## 5. Strength and Limitations

The strengths of this study included following a systematic approach to literature searching utilising the PRISMA-ScR guidelines, and investigated a range of business, marketing and life science databases, as well as grey literature. This provided holistic coverage of different perspectives on changing package sizes in both health and industrial sectors. All data were double screened, ensuring the reliability and consistency of results. Limitations of the study included the lack of foodservice data for the savoury snack category on Euromonitor, which may have been able to provide further insight into consumer purchasing behaviour of savoury snacks. Using individual package sizes instead of the size bands in this study may have been able to detect more detailed trends in changing package sizes. The CAGR measurement only captured overall growth or decline between two time points and may have missed more subtle changes within this period. Accessibility of other marketing and business databases such as IBISWorld were limited due to costs required to purchase reports, and industry perspectives from Japan and Hong Kong may have been missed if they were not available in English language. Publication bias is another possible limitation of this study as companies may publish data selectively for promotional purposes and reputational enhancement.

## 6. Conclusions

Quantifying the changes in package sizes of discretionary foods and beverages over time as well as understanding the industry reasons behind these changes is important for the development of public health policy to improve the food environment. This study showed that although savoury snacks are increasing in popularity, the growing sales of smaller package sizes is encouraging. This controlled portion size can potentially reduce energy intake from discretionary foods. By providing a wide range of smaller package size options in an acceptable price range, food manufacturers can help create a better food environment and can nudge consumers towards more appropriate portion size selections. Future studies can evaluate the effectiveness of this approach and its impact on total energy intake.

## Figures and Tables

**Figure 1 ijerph-19-09359-f001:**
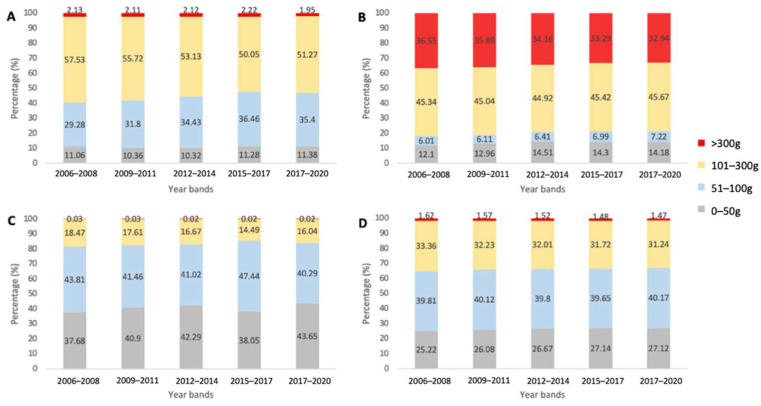
Proportion of total sales per capita of savoury snacks over three-year periods from 2006 to 2020 according to package size bands in (**A**) Australia, (**B**) the USA, (**C**) Japan, and (**D**) Hong Kong.

**Figure 2 ijerph-19-09359-f002:**
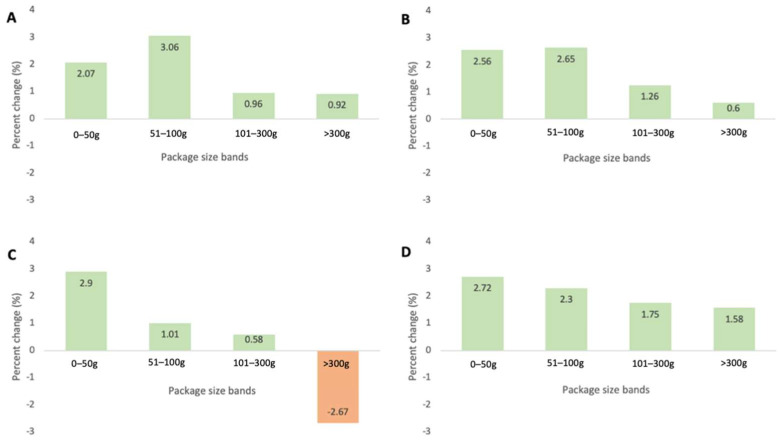
Compound annual growth rate (CAGR) per capita of savoury snack total sales from 2006 to 2020 according to package size bands in (**A**) Australia, (**B**) the USA, (**C**) Japan, and (**D**) Hong Kong.

**Figure 3 ijerph-19-09359-f003:**
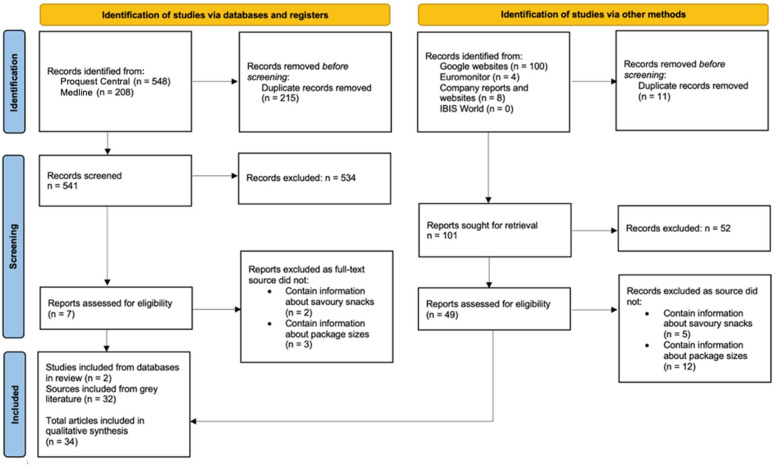
PRISMA flow chart of studies and other literature sources on savoury snacks included in the scoping review.

**Table 1 ijerph-19-09359-t001:** Total sales per capita of savoury snacks over three-year periods and proportion of increase from 2006 to 2020 in Australia, the USA, Japan, and Hong Kong.

	2006–2008	2009–2011	2012–2014	2015–2017	2018–2020	Proportion of Increase over Time, %
**Australia**	229.66	239.93	259.36	277.31	286.33	24.68
**USA**	172.92	174.96	184.66	189.32	201.14	16.32
**Japan**	229.64	244.09	254.11	267.35	280.7	22.23
**Hong Kong**	64.49	73.59	79.65	83.76	86.45	34.05

**Table 2 ijerph-19-09359-t002:** Industry reasons for changes in savoury snack package sizes.

	Increase in Package Size	Decrease in Package Size
**Country/region**		
USA	6	14
Australia	1	4
Hong Kong	0	1
Japan	0	3
Global	2	2
Reasons ^1^		
Health consciousness trends	0	15
Convenience	1	9
Portion control	0	8
COVID-19 snacking behaviour	6	2
Value for money	5	0
Increasing profit and product line expansion	1	4
Social sharing behaviour	3	0
Standardizing production and costs	0	3
International recommendations ^2^	0	2
Snacking as meal replacements	1	2
Affordability issues	0	2
Corporate responsibility	0	1

^1^ Multiple reasons may appear in one source. ^2^ By government bodies and the World Health Organization (WHO).

## Data Availability

Not applicable.

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
