# Peer review of "Changes in Package Sizes of Savoury Snacks through Exploration of Euromonitor and Industry Perspectives"

_ijerph, 2022, doi:10.3390/ijerph19159359_

Round 1
Reviewer 1 Report
1) The Introduction need to be revised.
2) Some sentences are confusing and vague. Please check it.
3) What is the current trends in sales data of savoury snacks according to package size? Justify.
4) The conclusion is too short.
5) The citation is not based on journal/articles.
6) Follow MDPI guidelines, especially for references.
Author Response
Thank you for your review of this manuscript, please see the attached document.

Reviewer 2 Report
This paper is a systematic review well organized following the PRISMA guidelines. It entitles on the changes in package sizes of savoury snacks through monitoring reliable and up-to-date data sources in the last 15 years. The results are interesting and of use for industry and marketing perspectives.
I have just one question for the authors. In several parts of the manuscript the authors used the word “significantly” in comparing sizes, consumption, and different types of snacks. Are these changes statistically significant? Could the authors compare the data that they obtained to see if they get values that are significantly different? (i.e., p < 0.05). This would further strengthen the results of the systematic review.
Regarding limitations, one of the most common limitations for systematic reviews is the publication bias which is something the authors could add to their limitations section. Consumption can also be under/over-reported which is part of publication bias.
Author Response

(The authors gave the same response as above.)

Reviewer 3 Report
This is an interesting manuscript. It is not earth-shattering in its methods, results or conclusions, but it does give one something to think about.
The producer economic factor mixed with consumer inability to remember package sizes is not dealt with in a major way here. I know that your literature did not likely cover this subject. However, producers often reduce the contents of a product as a means of dealing with inflationary pressures, and consumers are unlikely to perceive minor changes. It is true that portion control is an increasing factor in package sizes. It is also true that producers dropping sizes from 1.25 ounces to 1.00 ounce to increase profits or reduce costs is a factor, and consumers are unlikely to notice the size difference.
The paper needs some editing for word choice, grammar, readability. The sense of the article and its meaning is excellent. However, throughout the article single words (like "the" or "an") are missing which a native English reader will notice. The meaning of the sentence is understandable, but the flow of the sentence is broken. It just needs a good editor to help.
I enjoyed reading this. Interesting.
Author Response
Thank you for your review of this manuscript, please see the document attached,

Reviewer 4 Report
Thanks to the authors of the manuscript for their contribution to research related to the study of consumer buying habits and food producers in different countries.
Some comments and suggestions:
Figure 1 and Figure 2 – the same information is given. I would recommend leaving only the information given in Figure 2, as it better reflects the obtained data. It is also necessary to display the data using another type of graph, or to give the data in a table, because the information in this type of graph is incomprehensible / difficult to perceive.
Please add to the discussion on the results obtained using the PRISMA method.
Author Response

(The authors gave the same response as above.)

Round 2
Reviewer 1 Report
Overall is good and satisfactory.